# Fatigue Limit of Custom 465 with Surface Strengthening Treatment

**DOI:** 10.3390/ma13010238

**Published:** 2020-01-06

**Authors:** Gang An, Ren-jing Liu, Guang-qiang Yin

**Affiliations:** 1School of Management, Xi’an Jiaotong University, Xi’an 710049, China; an_gang@163.com (G.A.);; 2Qing’an Group Corporation Limited, Xi’an 710077, China

**Keywords:** nitriding, shot peening, fatigue performance, residual stress, hardness

## Abstract

In order to study the effect of nitriding or shot peening on the surface modification and fatigue properties of martensitic stainless-steel Custom 465, the residual stress and micro-hardness of the strengthened layer are determined by X-ray and micro-hardness tester, respectively. The up-and-down method is used to measure the rotational bending fatigue strength at 1 × 10^7^ cycles, and the fatigue fracture characteristic is observed by scanning electron microscopy. The relationship between surface residual stress and internal fatigue limit of surface strengthening treatment is discussed. Results show that nitriding or shot peening surface strengthening layer forms a certain depth of compressive residual stress, where in the surface compressive residual stress of the nitrided specimens is greater than the shot peened specimens. The micro-hardness of the nitrided or shot peened surface strengthening layer is significantly improved, where in the surface micro-hardness of nitriding specimens are higher than shot peening specimens. The nitriding or shot peening surface strengthening can significantly improve the fatigue limit of Custom 465, wherein the fatigue limits of nitrided and shot peened surface strengthened specimens are 50.09% and 50.66% higher than that of the un-surface strengthened specimens, respectively. That is, the effect of the two strengthening methods on fatigue limit is not very different. The fracture characteristics show that the fatigue crack of the un-surface strengthened specimens originates from the surface, while the fatigue crack of surface strengthened specimens originates from the subsurface layer under the strengthened layer. The relationship between the internal fatigue limit and the surface residual stress of the surface strengthened specimen can be used as a method for predicting the fatigue limit of the surface strengthened specimens.

## 1. Introduction

Custom 465 precipitation-hardened martensitic stainless steel, which has both well mechanical properties of martensitic time-effective steel and corrosion resistance of stainless steel [1]. However, the rotational bending fatigue limit of this type of steel was much lower than half of the tensile strength [2]. Therefore, how to improve its fatigue performance is worth further study.

Surface strengthening treatment is the main method to improve the fatigue strength of the material. It is to establish compressive residual stress and harden the surface layer. Surface strengthening treatment methods used in engineering include surface heat treatment, surface chemical heat treatment, and surface coating strengthening [3]. Surface heat treatment includes flame quenching and induction quenching. Surface chemical heat treatment includes nitriding, carburizing and carbonitriding. Surface coating strengthening includes shot peening, rolling pressure, hammer and overload stretching. Shot peening is used as a pneumatic machine to the surface of the parts, so that it produces plastic deformation and compressive residual stress, thereby increasing fatigue strength. It is the most widely used. Torres et al. [4] researched the fatigue life of four shot peening conditions for AISI 4340 steel and the compressive residual stress field was measured by an X-ray diffraction apparatus during fatigue tests, in addition, the fracture surfaces were analyzed using a scanning electron microscope, and pointed out that the evaluation of fatigue life, relaxation of compressive residual stress field and crack sources. Segurado et al. [5] studied the fatigue life increased after quenched and tempered AISI4340 steel by shot peening, and pointed out the effects of fatigue life for using different types of shots, intensities and coverages. Trung et al. [6] researched the effects of the process conditions for both single and double shot peening on the fatigue life of AISI 4340 steel. Trung et al. [7] studied the effect of Inconel 718 tested in low and high cycle fatigue by different shot peening conditions, and analyzed the relationship between the fatigue life and crack initiation mechanisms, residual stress relaxation, process induced strain hardening and surface roughness. As a chemical heat treatment technology, nitriding technology can improve the surface hardness, fatigue strength and wear resistance of parts, and thus improve the service life of parts. Therefore, it has been widely used in the industrial field. Nitrification is the placement of parts into nitrogenous media and heating of 10 min–100 h at 500–600 °C. Nitrogen atoms in the medium penetrate into the surface of the parts. With the alloy composition in the steel such as AI, Cr, Mo, V, W, Ti, etc., a diffuse distribution of nitrogen in the surface (650–1000 HV) is formed, and its volume is expanded, resulting in compressive residual stress in the surface layer. Thus, the wear resistance and fatigue strength of the surface layer are improved. Guagliano [8] researched an approach based on fracture mechanics concepts to assess the influence of shot peening on nitrided components, and proposed to determine the threshold values of the stress intensity factors of the nitrided and shot peened components. Pariente et al. [9] studied the fatigue performance of the nitrided 42CrMo4 steel by shot-peening and determined the micro-structure, the micro-hardness, the residual stresses distribution and the crack resistance. Croccolo et al. [10] designed three-level experiments by the Design of Experiment were conducted on un-notched and notched nitrided and shot-peened 32CrMoV13 steel, and determined the optimal nitriding and shot peening process parameters of the maximal fatigue strength.

In this paper, the residual stress and micro-hardness distribution of the strengthened layer under the condition of shot peening and nitriding are determined by X-ray and micro-hardness apparatus, respectively, in order to reveal the effect of surface strengthening treatment on surface modification. The rotational flexural fatigue strength of the specimens subjected to surface strengthening treatment, shot peening and nitriding under 1 × 10^7^ cycles is determined by the up-and-down method, and the characteristics of the fatigue fracture are observed by scanning electron microscope to reveal the effects of surface strengthening treatment in the fatigue performance of Custom 465. Finally, the relationship between the internal fatigue limit and the surface residual stress of the surface strengthened specimen is discussed.

## 2. Materials and Experiments

### 2.1. Materials

The material considered in this work is the Martensitic aging stainless steel Custom 465, the composition of this steel (wt%) is determined by Inductively Coupled Plasma Atomic Emission Spectroscopy with ICPE-9000 analysis (Shimadzu, Kyoto, Kyoto Prefecture, Japan) as: 0.0068 C, 10.97 Cr, 11.10 Ni, 0.95 Mo, 1.43 Ti, and others Fe. The room temperature mechanical properties measured results are: ultimate tensile strength σ_b_ = 1700 MPa, initial yield strength σ_0.2_ = 1560 MPa, tensile elongation A = 13.5%, cross-section shrinkage rate <S> = 63.5%, and hardness of HV ≥ 471. These properties are obtained by means of the heat treatment process: (a) Solutionizing heat treatment—Heat to 900 ± 10 °C, holding for one hour, and followed by oil cooling; (b) Cold-treating —For optimum aging response, solution annealing should be followed by refrigerating to −73 °C, holding for four hours, then warming to room temperature; (c) Aging heat treatment—Heat to 520 ± 5 °C, holding for four hours, and followed by air cooling. The surface treatment is commonly used in the industry to mechanical components fatigue properties, this study aims to quantify the effects of nitriding or shot peening after heat treatment process used in aerospace industry of Custom 465 in high cycle fatigue at room temperature. The surface treatment of material involves as follows: (a) Nitriding—The material was nitrided at 510 °C in a well furnace with a depth of 0.15–0.25 mm and a final surface hardness of HV ≥ 746. (b) Shot peening—The material was shot peened using cast steel balls with S110 and a hardness of HV595–832 under an Almen intensity of 0.2–0.3A, and coverage of 200% were applied when all shot peening procedures in a pneumatic shot peening equipment. After grinding and polishing, the specimen of the gold phase was corroded by ferric chloride solution and the microstructure of the gold phase was examined under a model Axio Observer A1m Zeiss (Zeiss, Oberkochen, Baden-Württemberg, Germany) Inverted Metallurgical Microscope. Figure 1 presents the microstructure of the different surface treatments involve: (a) microstructure of the un-surface strengthened (only heat treatment) specimen, (b) surface microstructure of the shot peened (after heat treatment) specimen and (c) nitriding layer microstructure of the nitrided (after heat treatment) specimen.

### 2.2. Experiments

The residual stresses distribution along depth was determined by a model Xstress-3000 X-ray diffraction apparatus (Stresstech Oy, Somija, Vaajakoski, Finland) combined with the layer by layer polishing method, residual stresses along depth were measured every 50 μm, three points were measured at the depth of each layer, and the mean value was used as the residual stress of the depth of the layer. The micro-hardness was measured by a model Tukon-1102 digital display micro-hardness apparatus (Wilson, Lake Bluff, IL, USA). Rotating bending fatigue tests were performed in a 3000 rpm by a model PQ1-6 bending fatigue machine (Qingshan Testing Machine Factory, Yinchuan, China). Three types of specimens were used to perform fatigue tests: (a) un-surface strengthened (only heat treatment) specimen, (b) shot peened (after heat treatment) specimen and (c) nitrided (after heat treatment) specimen. The fatigue specimen geometry is shown in Figure 2. Test environment conditions: temperature (25 ± 2) °C, humidity (25–40%)RH, constant amplitude load, stress ratio of R = −1, the up-and-down method to determine the fatigue limit at a specified life of 1 × 10^7^ cycles. A model Tescan scanning electron microscopy (SEM) was used to analyze the fracture of the specimen with the stress level slightly higher than the fatigue limit to determine the position and characteristics of the fatigue crack source.

## 3. Results and Discussion

### 3.1. Residual Stress

Residual stress distributions along the depth of the layer after shot peened or nitrided of the specimens determined by X-ray are shown in Figure 3. It showed the surface residual stress of the shot peened or nitrided strengthened specimen was compressive stress, in which the surface compressive residual stress of nitrided specimen was greater than shot peened specimen.

### 3.2. Micro-Hardness

The micro-hardness distribution along the depth of the layer of the Custom 465 specimen determined by a micro-hardness apparatus and strengthened by shot peened or nitrided is shown in Figure 4. It showed the micro-hardness of the shot peened or nitrided strengthened specimen was higher than that of the core, and the surface micro-hardness of the nitrided specimen was higher than that of the shot peened specimen.

### 3.3. Fatigue Limit

#### 3.3.1. Test Results

The determination of the fatigue limits of un-surface treatment, shot peened and nitrided specimens under the specified fatigue life 1 × 10^7^ cycles of material by the up-and-down method are shown in Figure 5.

From the up-and-down fatigue strength data of Figure 5, according to statistical theory, it can be concluded that the median fatigue limits were 529 MPa, 797 MPa and 794 MPa un-surface strengthening treatment, shot peened and nitrided specimens for 1 × 10^7^ cycles, respectively. Using un-surface strengthened specimens as the reference point, the fatigue limit of Custom 465 were increased by 50.66% and 50.09% by shot peened and nitrided specimens, respectively.

The typical fatigue fracture morphologies of the un-surface strengthening treatment, shot peened and nitrided Custom 465 observed with scanning electron microscopy are shown in Figure 6a–c, respectively. As can be seen from Figure 6, the fatigue crack source of the specimen without surface strengthening treatment was located on the surface, and the fatigue crack source of the surface strengthening treatment specimen was located under the surface strengthened layer or in the surface strengthen layer.

The fatigue test showed that the un-surface strengthening treatment specimen, the fatigue source was located on the surface of the specimen, and the fatigue limit was the surface fatigue limit, 529 MPa. The surface strengthened specimen and the fatigue crack were initiated in the Compressive residual stress filed below surface compressive residual stress layer. The fatigue source depth of shot peened specimen and nitrided specimen were about 300 μm, and the apparent fatigue limits were 797 MPa and 794 MPa, respectively. The experimental results that under the condition of R = −1, the crack was born in the Compressive residual stress filed below the compressive residual stress layer under the appropriate surface strengthening treatment, but the apparent fatigue limit was significantly increased. It was shown that the critical resistance of fatigue crack initiation in specimen was higher than that of un-surface strengthened specimen initiation on surface due to surface strengthening treatment.

#### 3.3.2. Predictive Results

The critical condition for the initiation of a fatigue crack inside the specimen is that the total of the residual stress and the load stress in a certain area is equal to the internal fatigue limit.
(1)$$\sigma_{i\left(-1\right)}=\sigma_{ri}+\sigma_{li}$$

In Equation (1), $\sigma_{i\left(-1\right)}$ is the internal fatigue limit, MPa; $\sigma_{ri}$ is the residual stress, MPa; and $\sigma_{li}$ is the load stress, MPa.

The internal fatigue source was located near the peak of compressive residual stress filed, the empirical expression of tensile residual stress field was given [11].
(2)$$\sigma_{rt}\left(Z\right)=\frac{{\left(Z-Z_{0}\right)}^{1.35}}{a{\left(Z-Z_{0}\right)}^{2}+b}$$

In Equation (2), *Z* is the depth from the surface, μm; $\sigma_{rt}\left(Z\right)$ is the tensile residual stress at *Z*, μm; $Z_{0}$ is the depth of the tensile residual stress field, MPa; *a*, *b* are the constant, and the available Equation (3) was determined.
(3)$$\frac{Z_{mt}-Z_{0}}{Z_{0}}=0.23$$

In Equation (3), $Z_{mt}$ is the depth at the peak of the tensile residual stress, μm.
(4)$$-\int_{0}^{Z_{0}}\left(r-Z\right)\sigma_{rc}\left(Z\right)dz=\int_{Z_{0}}^{r}\left(r-Z\right)\sigma_{rt}\left(Z\right)dz$$

In Equation (4), *r* is the radius of the specimen, mm; and $\sigma_{rc}\left(Z\right)$ is the compressive residual stress at depth *Z*, μm.

During the cyclic loading process, the local compressive stress in the surface strengthened specimen will exceed the yield limit, resulting in local yield, which will change residual stress distribution, that was, static load relaxation. Wangrenzhi [12] and Qiuqiong [13] the law of residual stress static loading relaxation was studied. Relaxation residual stress was calculated by Equation (5).
(5)$$\sigma_{r}^{\prime }=\sigma_{-s}^{*}-\sigma_{a}$$

In Equation (5), $\sigma_{r}^{\prime }$ is the relaxation residual stress, MPa; $\sigma_{-s}^{*}$ is the equivalent compression yield strength, MPa; and $\sigma_{a}$ is the cyclic stress amplitude, MPa. For Custom 465 steel, $\sigma_{-s}^{*}$ is close to $\sigma_{0.2}$, circulating under apparent fatigue limit, there was cyclic stress amplitude.
(6)$$\sigma_{a}=\sigma_{ap\left(-1\right)}\left(1-\frac{Z}{r}\right)$$

In Equation (6), $\sigma_{ap\left(-1\right)}$ is the apparent fatigue limit of fatigue source inside, MPa.

The tensile residual stress distribution was calculated from compressive residual stress field of surface strengthened specimen after cycle stress, and it described in Figure 7. The peak values of tensile residual stress were about 175 MPa and 171 MPa, respectively, and depth was about 350 μm, which was consistent with fatigue source depth. Therefore, tensile residual stresses at fatigue source of shot peened and nitrided specimens were approximately 175 MPa and 171 MPa, respectively.

Load stress at fatigue source was calculated by Equation (7).
(7)$$\sigma_{li}=\left(1-\frac{Z_{mt}}{r}\right)\sigma_{ap\left(-1\right)}$$

In Equation (7), $\sigma_{li}$ is the load stress at fatigue source inside, MPa.

Substituting $\sigma_{ri}$ and $\sigma_{li}$ into the Equation (1), the stress of fatigue crack initiation inside specimen can be obtained, that is, the internal fatigue limit. The predictive results of material’s fatigue limit under different surface treatments are listed in Table 1.

Therefore, the internal fatigue limit of Custom 465 of the shot peened specimen was 879 MPa, which was 10.3% different from the test value of 797 MPa; the internal fatigue limit of Custom 465 of the nitrided specimen was 876 MPa, which is 10.4% different from the test value of 876 MPa.

## 4. Conclusions

(1)Shot peened or nitrided surface strengthening treatment form a favorable compressive residual stress and micro-hardness distribution state on the surface of Custom 465.(2)Shot peened or nitrided surface strengthening treatment can significantly increase the fatigue limit of Custom 465, and the effect of nitrided and shot peened increase the fatigue limit was not significant, that was, about 50%.(3)The fatigue crack source of the un-surface strengthened specimen is located on the surface, while the fatigue crack source of the strengthened specimen is located on the sub-surface under the strengthened layer.(4)The relationship between the internal fatigue limit and the surface residual stress of the surface strengthened specimen can be used as a method for predicting the fatigue limit of the surface strengthened specimens.

## Figures and Tables

**Figure 1 materials-13-00238-f001:**
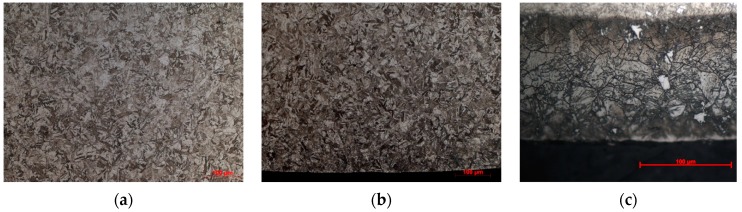
Microstructure of the surface under different treatments (**a**) un-surface treatment specimens; (**b**) shot peened specimens; (**c**) nitrided specimens.

**Figure 2 materials-13-00238-f002:**
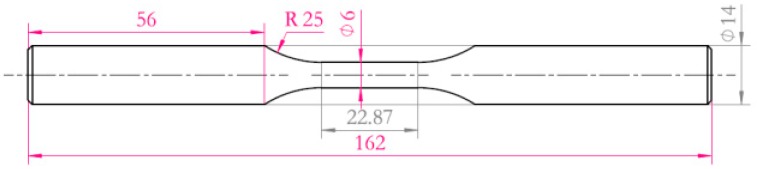
Shape and size of the specimens used for fatigue tests.

**Figure 3 materials-13-00238-f003:**
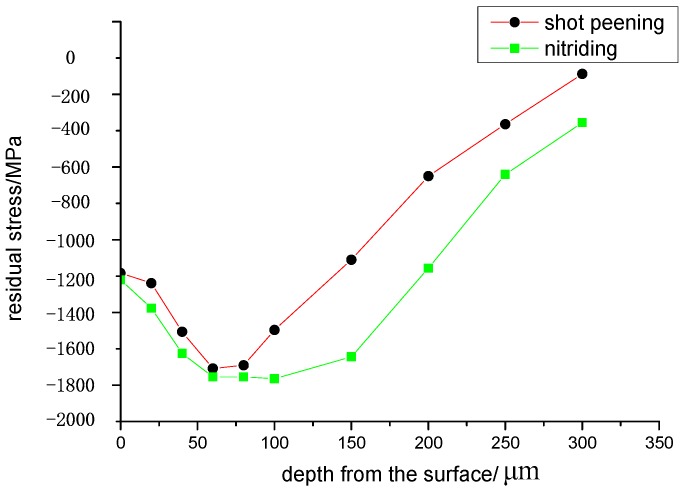
Residual stress distribution under different surface treatments.

**Figure 4 materials-13-00238-f004:**
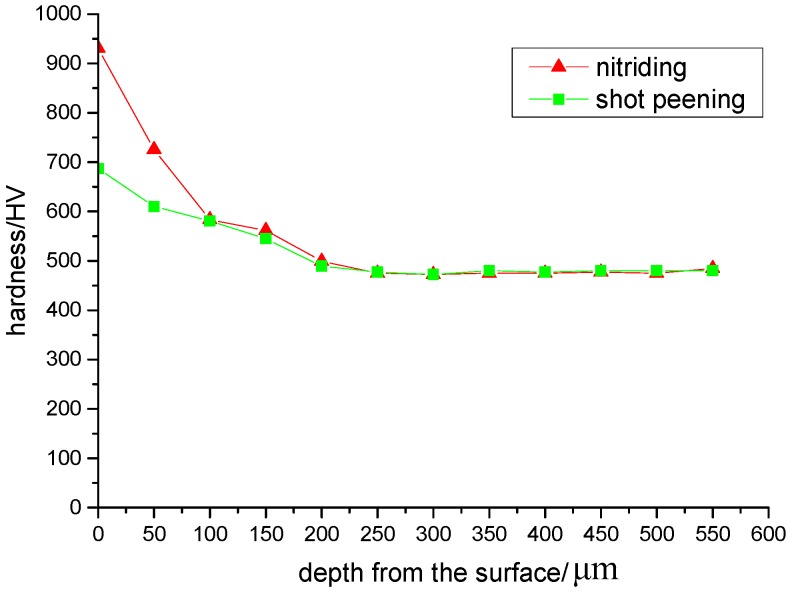
Micro-hardness distribution under different surface treatments.

**Figure 5 materials-13-00238-f005:**
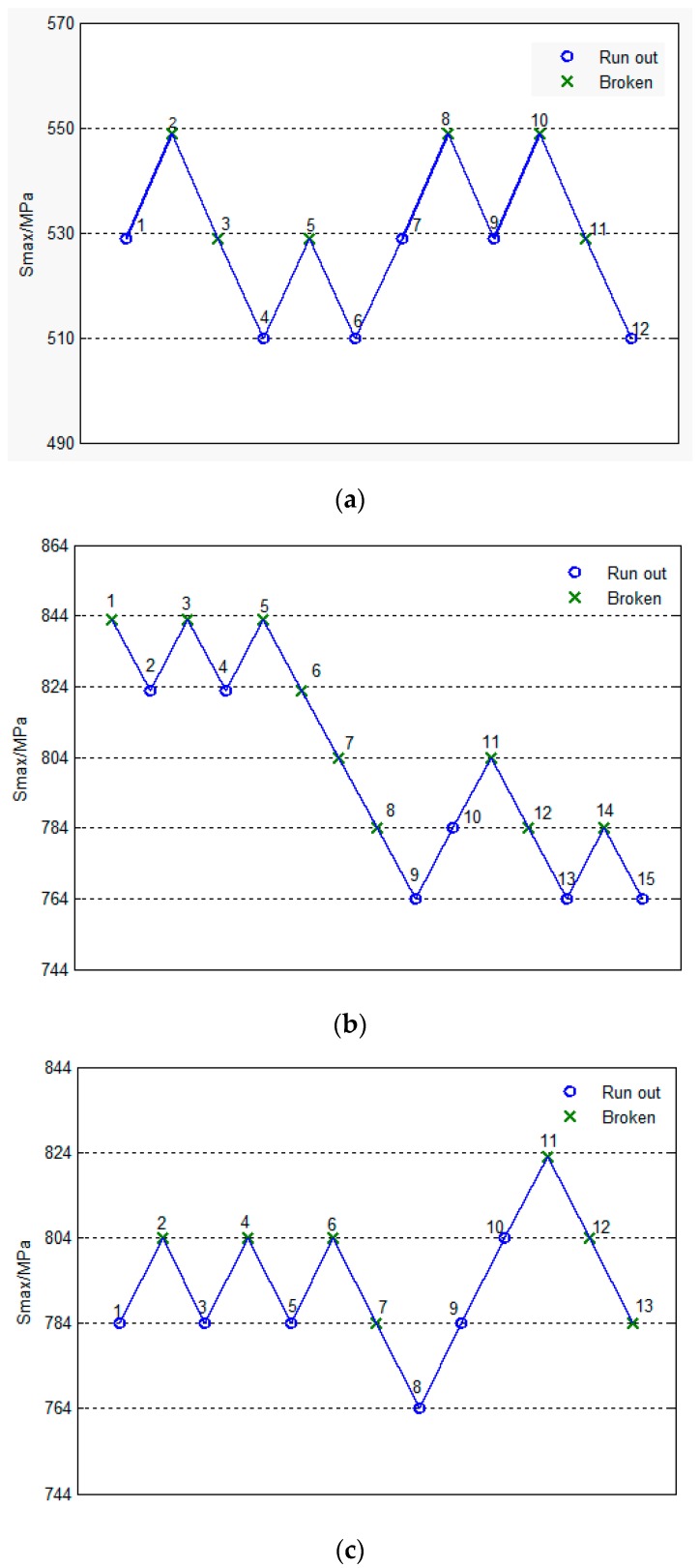
The up-and-down map of fatigue tests (**a**) un-surface treatment specimens; (**b**) shot peened specimens; (**c**) nitrided specimens.

**Figure 6 materials-13-00238-f006:**
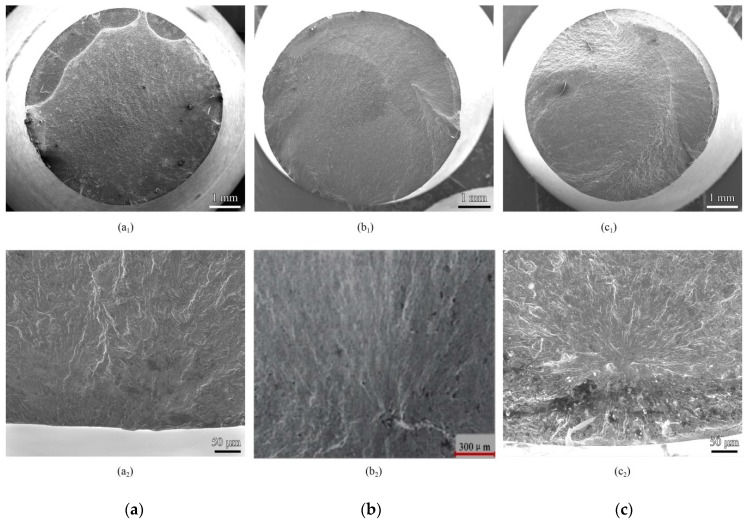
Fractograph of fatigue fracture (**a**) Un-surface strengthened specimen; (**b**) Shot peened specimen; (**c**) Nitrided specimen.

**Figure 7 materials-13-00238-f007:**
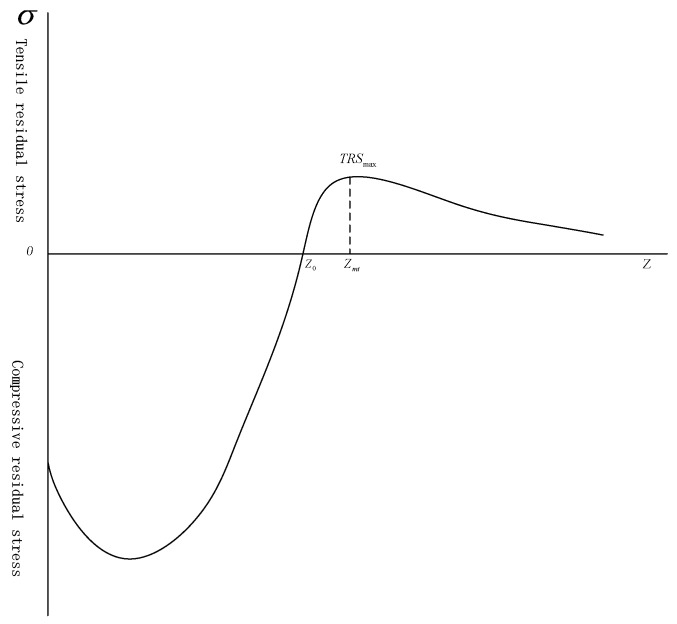
Tensile residual stress field of surface strengthened specimen after cycle stress.

**Table 1 materials-13-00238-t001:** Fatigue limit of predictive and test results under different surface treatments.

Specimen	Fatigue Limit		Deviation
	Test Results	Predictive Results	
Un-surface strengthened	529 MPa	-	-
Shot peened	797 MPa	879 MPa	10.3%
Nitrided	794 MPa	876 MPa	10.4%

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
