# Peer review of "Fatigue Limit of Custom 465 with Surface Strengthening Treatment"

_materials, 2020, doi:10.3390/ma13010238_

Round 1

Reviewer 1 Report

Dear Authors,

few corrections in the article should be given:

Different types of names for untreated surface is given in the text (non-surface-strengthened, un-sufrace). I recomend to uniform it, and use one term in whole article, also in the Abstract and under the figures. Is the un-surface strengthened sample after heat treatment? It should be given clearly in the "Materials" part. Line 40 - method "carburizing" is given two times. Should it be other method given here? Line 59 - aluminium (AL) should be given by Al. Line 60 - "of nitrogen in the surface", instead of "nitride surface". Line 63 - "nitrided" instead of "nitride". Line 82 - how the chemical composition was measured? The method and apparatus should be given. Line 85 and 86 - what "OC" and "AC" means? Line 88 - hardness in Vickers scale is much easier to understand than HR15N. I recommend to use Vickers scale, as it is shown in Fig. 4. Line 93 - "Fig. 1 presents the microstructure", instead of "was". Line 98 - Description under the figure 1 should be changed as follow "Microstructure of the surface under different treatments". Is the magnification of each figure the same? Maybe it will be better to show the microstructure of the sample (a) and (b) near to the surface like in Fig. 1 (c). Line 107 - "is shown", instead of "was shown". The name of X-ray and SEM aparatus could be given. Line 117 - "are shown", instead of "was shown". Line 124 - "is shown", instead of "was shown". Figures 5, 6 and 7 is better to made as one Figure with a, b and c options. Figure 8 b2 is really poor quality. It have to be corrected. Are the parameters Z0, Zmt, a and b in equation 2, 3 and 4 read from the figure 3? How it was estimate? Could it be marked on Fig 3 by dot line? Line 203 - "shot", instead of "Shot". A bar graph or at least summary table for "Predictive results" should be given. Showing results in the text is hard to read.

Author Response

Dear Editors and Reviewers:

Thank you very much for your careful review and constructive suggestions with regard to our manuscript “Fatigue limit of Custom 465 with Surface Strengthening Treatment” (ID: 672651). Those comments are all valuable and very helpful for revising and improving our paper. We have studied comments carefully and have made correction which we hope meet with approval. Revised portion is marked in red in revised manuscript. The main corrections in the manuscript and the responds to the reviewer’s comments are as flowing:

Different types of names for untreated surface is given in the text (non-surface-strengthened, un-surface). I recommend to uniform it, and use one term in whole article, also in the Abstract and under the figures.

Response: Thanks. “non-surface-strengthened, un-surface strengthened and no-surface strengthened” were unified as “un-surface strengthened” in revised manuscript.

Is the un-surface strengthened sample after heat treatment? It should be given clearly in the "Materials" part.

Response: Yes. It had been given in the "Materials" part in revised manuscript.

For example, “The surface treatment is commonly used in the industry to mechanical components fatigue properties, this study aims to quantify the effects of nitriding or shot peening after heat treatment process used in aerospace industry of Custom 465 in high cycle fatigue at room temperature.”

Line 40 - method "carburizing" is given two times. Should it be other method given here?

Response: Thanks. The second “carburizing” was corrected as “carbonitriding” in revised manuscript.

Line 59 - aluminium (AL) should be given by Al.

Response: Thanks. The “AL” was corrected as “Al” in revised manuscript.

Line 60 - "of nitrogen in the surface", instead of "nitride surface".

Response: Thanks. The “nitride surface” was corrected as “of nitrogen in the surface” in revised manuscript.

Line 63 - "nitrided" instead of "nitride".

Response: Thanks. The “nitride” was corrected as “nitrided” in revised manuscript.

Line 82 - how the chemical composition was measured? The method and apparatus should be given.

Response: Agreed. The method and apparatus of the chemical composition were given in revised manuscript.

For example, “the composition of this steel (wt%) is determined by Inductively Coupled Plasma Atomic Emission Spectroscopy with ICPE-9000 analysis as”

Line 85 and 86 - what "OC" and "AC" means?

Response: The “OC” means “oil cooling”, and the “AC” means “air cooling” were corrected in revised manuscript.

For example, “These properties are obtained by means of the heat treatment process: (a) Solutionizing heat treatmen — Heat to 900±10°C, holding for one hour, and followed by oil cooling; (b) Cold-treating — For optimum aging response, solution annealing should be followed by refrigerating to -73°C, holding for four hours, then warming to room temperature; (c) Aging heat treatmen — Heat to 520±5°C, holding for four hours, and followed by air cooling.”

Line 88 - hardness in Vickers scale is much easier to understand than HR15N. I recommend to using Vickers scale, as it is shown in Fig. 4.

Response: Agreed. The “HR15N≥91” was corrected as “HV≥746” in revised manuscript.

Line 93 - "Fig. 1 presents the microstructure", instead of "was".

Response: Thanks. The “was” was corrected as “Fig. 1 presents the microstructure” in revised manuscript.

Line 98 - Description under the figure 1 should be changed as follow "Microstructure of the surface under different treatments". Is the magnification of each figure the same? Maybe it will be better to show the microstructure of the sample (a) and (b) near to the surface like in Fig. 1 (c).

Response: Agreed. The “Surface microstructure of under different surface treatments” was changed as “Microstructure of the surface under different treatments” and the “Fig.1”has been replaced in revised manuscript.

For example,

"Figure 1. Microstructure of the surface under different treatments (a) un-surface treatment specimens; (b) shot peened specimens. (c) nitrided specimens."

Line 107 - "is shown", instead of "was shown".

Response: Agreed. The “was shown” was corrected as “is shown” in revised manuscript.

The name of X-ray and SEM apparatus could be given.

Response: Agreed. The name of X-ray and SEM apparatus had been given in revised manuscript.

For example, “The residual stresses distribution along depth was determined by a model Xstress-3000 X-ray diffraction apparatus combined with the layer by layer polishing method, residual stresses along depth were measured every 50μm, three points were measured at the depth of each layer, and the mean value was used as the residual stress of the depth of the layer. The micro-hardness was measured by a model Tukon-1102 digital display micro-hardness apparatus. Rotating bending fatigue tests were performed in a 3000rpm by a model PQ1-6 bending fatigue machine. Three types of specimens were used to perform fatigue tests: (a) un-surface strengthened (only heat treatment) specimen, (b) shot peened (after heat treatment) specimen and (c) nitrided (after heat treatment) specimen. The fatigue specimen geometry is shown in Figure 2. Test environment conditions: temperature (25±2)°C, humidity (25%~40%)RH, constant amplitude load, stress ratio of R=-1, the up-and-down method to determine the fatigue limit at a specified life of 1×107 cycles. A model Tescan scanning electron microscopy (SEM) was used to analyze the fracture of the specimen with the stress level slightly higher than the fatigue limit to determine the position and characteristics of the fatigue crack source.”

Line 117 - "are shown", instead of "was shown".

Response: Thanks. The “was shown” was corrected as “are shown” in revised manuscript.

Line 124 - "is shown", instead of "was shown".

Response: Thanks. The “was shown” was corrected as “is shown” in revised manuscript.

Figures 5, 6 and 7 are better to made as one Figure with a, b and c options.

Response: Agreed. The Figures 5, 6 and 7 were made as one Figure with a, b and c options in revised manuscript.

Figure 8 b2 is really poor quality. It has to be corrected.

Response: Agreed. The “Figure 8 b2” has been corrected in revised manuscript.

Are the parameters Z0, Zmt, a and b in equation 2, 3 and 4 read from the figure 3? How it was estimate? Could it be marked on Fig 3 by dot line?

Response: The “Figure 7” has been added in revised manuscript.

For example, “The parameters Z0, Zmt are shown in Figure 7.

Figure 7. Tensile residual stress field of surface strengthened specimen after cycle stress.”

Line 203 - "shot", instead of "Shot".

Response: Thanks. The “Shot” was corrected as “shot” in revised manuscript.

A bar graph or at least summary table for "Predictive results" should be given. Showing results in the text is hard to read.

Response: Agreed. The “Table 1” has been added in revised manuscript.

For example, “The predictive results of material’s fatigue limit under different surface treatments are listed in Table 1.

Table 1. Fatigue limit of predictive and test results under different surface treatments.

Specimen

Fatigue limit

Deviation

Test results

Predictive results

Un-surface strengthened

529MPa

-

-

Shot peened

797MPa

879MPa

10.3%

Nitrided

794MPa

876MPa

10.4%

We tried our best to improve the manuscript and marked in red in revised manuscript. We appreciate for Editors/Reviewers’ warm work earnestly, and hope that the correction will meet with approval.

Reviewer 2 Report

Dear Sir

The main defect of the paper is that all of the measurements are not used for the quantitative approach. In the chapter 3.2.2 is necessary to use the offered formulas for calculation of the values of the experiment (now this chapter is offered at theoretical level of how must be worked the measurement values).  The measured values must be put in the formulas and the conclusions must be quantitative instead to be qualitative.

Some small corrections:

 Line 40,41: Appears two times "carburizing". Please do the correction!

Line 88 -After HR15N>91 I consider that is needed to put a point and start the new phrase: "..The shot peening...".

Line 93 and fig.1 - MUst to be given the scanning microscope type and used parameters. Also in the fig 1 these parameters must be mentioned.

Line 119: Tesidual stress of nitriding specimens is greater than shot peened specimens in fig 4 not in the fig 3, as is written.

Fig.8 What means "high magnification"??? Please mention the values and the base of comparation.

Equations 1,2...etc. Please mention the measurement units of all of the parameters from these equations.

Author Response

Dear Editors and Reviewers:

Thank you very much for your careful review and constructive suggestions with regard to our manuscript “Fatigue limit of Custom 465 with Surface Strengthening Treatment” (ID: 672651). Those comments are all valuable and very helpful for revising and improving our paper. We have studied comments carefully and have made correction which we hope meet with approval. Revised portion is marked in red in revised manuscript. The main corrections in the manuscript and the responds to the reviewer’s comments are as flowing:

1. Line 40, 41: Appears two times "carburizing". Please do the correction!

Response: Thanks. The second “carburizing” was corrected as “carbonitriding” in revised manuscript.

Line 88 -After HR15N>91 I consider that is needed to put a point and start the new phrase: "..The shot peening...".

Response: Agreed. It was corrected in revised manuscript.

For example, “The room temperature mechanical properties measured results are: ultimate tensile strength σb=1700MPa, initial yield strength σ0.2=1560MPa, tensile elongation A=13.5%, cross-section shrinkage rate <S> = 63.5%, and hardness of HV≥471. These properties are obtained by means of the heat treatment process: (a) Solutionizing heat treatmen — Heat to 900±10°C, holding for one hour, and followed by oil cooling; (b) Cold-treating — For optimum aging response, solution annealing should be followed by refrigerating to -73°C, holding for four hours, then warming to room temperature; (c) Aging heat treatmen — Heat to 520±5°C, holding for four hours, and followed by air cooling. The surface treatment is commonly used in the industry to mechanical components fatigue properties, this study aims to quantify the effects of nitriding or shot peening after heat treatment process used in aerospace industry of Custom 465 in high cycle fatigue at room temperature. The surface treatment of material involves as follows: (a) Nitriding — The material was nitrided at 510°C in a well furnace with a depth of 0.15-0.25mm and a final surface hardness of HV≥746. (b) Shot peening — The material was shot peened using cast steel balls with S110 and a hardness of HV595~832 under an Almen intensity of 0.2~0.3A, and coverage of 200% were applied when all shot peening procedures in a pneumatic shot peening equipment.”

Line 93 and fig.1 - Must to be given the scanning microscope type and used parameters. Also in the fig 1 these parameters must be mentioned.

Response: Agreed. The apparatus type and used parameters had been given,and the Fig.1 has been corrected in revised manuscript.

For example, “After grinding and polishing, the specimen of the gold phase was corroded by ferric chloride solution and the microstructure of the gold phase was examined under a model Axio Observer A1m Zeiss Inverted Metallurgical Microscope.”

Line 119: Residual stress of nitriding specimens is greater than shot peened specimens in fig 4 not in the fig 3, as is written.

Response: It has been checked, yes, the residual stress of nitriding specimens is greater than shot peened specimens in the fig.3.

Fig.8 What means "high magnification"??? Please mention the values and the base of comparation.

Response: The “high magnification” meaningless, it has been deleted in revised manuscript.

Equations 1,2...etc. Please mention the measurement units of all of the parameters from these equations.

Response: Thanks. The measurement units of all the parameters were added in revised manuscript.

We tried our best to improve the manuscript and marked in red in revised manuscript. We appreciate for Editors/Reviewers’ warm work earnestly, and hope that the correction will meet with approval.

Round 2

Reviewer 1 Report

Dear Authors, I approve the introduced corrections.

Reviewer